# Effects of Low-Temperature Stress on Physiological Characteristics and Microstructure of Stems and Leaves of *Pinus massoniana* L.

**DOI:** 10.3390/plants13162229

**Published:** 2024-08-11

**Authors:** Hu Chen, Xingxing Liang, Zhangqi Yang

**Affiliations:** 1Guangxi Key Laboratory of Superior Timber Trees Resource Cultivation, Guangxi Forestry Research Institute, Nanning 530002, China; chenhubeijing-2008@163.com (H.C.); cindyliang26@163.com (X.L.); 2Key Laboratory of Central South Fast-Growing Timber Cultivation of Forestry Ministry of China, Masson Pine Engineering Research Center of the State Forestry Administration, Nanning 530002, China; 3Masson Pine Engineering Research Center of Guangxi, Nanning 530002, China

**Keywords:** *Pinus massoniana* L., low-temperature stress, physiological and biochemical indexes, microstructure of tissue

## Abstract

*Pinus massoniana* L. is one of the most important conifer species in southern China and is the mainstay of the forest ecosystem and timber production, yet low temperatures limit its growth and geographical distribution. This study used 30-day-old seedlings from families of varying cold-tolerance to examine the morphological traits of needles and stems, chlorophyll fluorescence characteristics, protective enzymes, and changes in starch and lignin under different low-temperature stresses in an artificial climate chamber. The results showed that the seedlings of *Pinus massoniana* exhibited changes in phenotypic morphology and tissue structure under low-temperature stress. Physiological and biochemical indexes such as protective enzymes, osmoregulatory substances, starch, and lignin responded to low-temperature stress. The cold-tolerant family increased soluble sugars, starch grain, and lignin content as well as peroxidase activity, and decreased malondialdehyde content by increasing the levels of actual photochemical efficiency (*ΦPSII*), electron transport rate (ETR), and photochemical quenching (*qP*) to improve the cold tolerance ability. This study provides a reference for the selection and breeding of cold-tolerant genetic resources of *Pinus massoniana* and the mechanism of cold-tolerance, as well as the analysis of the mechanism of adaptation of *Pinus massoniana* in different climatic regions of China.

## 1. Introduction

Low temperature is a significant ecological element that limits plant growth, development, and geographical distribution, and extreme low-temperature stress results in plant mortality [1,2]. Low-temperature stress can be divided into chilling injury (0–15 °C) and freezing injury (<0 °C) [3]. Plant morphology, tissue structure, cell membrane system, protective enzyme system, and physiological metabolism in plants under low temperature have been extensively studied in recent years. Additionally, the selection and breeding of new cold-resistant varieties have also been explored [4,5]. Plant cytoplasmic membranes, antioxidant enzyme systems, osmoregulatory substances, endogenous hormones, and other stress responses are formed in response to low-temperature stress, and defense is carried out by a complex regulatory system [6]. Plants respond to low-temperature stress by activating antioxidant enzyme systems to scavenge reactive oxygen species and by producing more osmoregulatory substances such as proline, soluble sugars (SS), and soluble proteins, which enhance their osmotic regulatory capabilities [7,8,9,10]. The cold-resistant family had lower cell membrane permeability, relative conductivity, and malondialdehyde (MDA) content [11,12]. Chlorophyll fluorescence metrics can adapt to the effects of adversities on photosynthesis [13,14]. Recent studies have applied chlorophyll fluorescence as an indicator of photosynthetic efficiency in response to cold and freezing [15]. Increasing lignin concentration has recently been discovered to be an essential way for plants to tolerate abiotic stressors [16,17]. Researchers have recently focused on the link between lignin and starch production pathways and cold tolerance [17,18]. Our previous physiological and molecular studies on the response of *Pinus massoniana* L. seedlings’ roots to low temperature, drought, and lead stress have preliminarily confirmed that the lignin synthesis pathway plays an important role in *P*. *massoniana*’s response to abiotic stress [19,20].

*P*. *massoniana*, which belongs to Pinaceae and genus *Pinus*, is one of the most important high-quality coniferous timber species in southern China [21]. It is fast-growing, productive, adaptable, and widely distributed, and has a high degree of comprehensive uses, a high economic value, and other excellent characteristics. Due to global climate variability and the increasing frequency of extreme weather events, *P. massoniana* production areas have experienced numerous incidents of rain, snow, freezing conditions, and other low-temperature disasters. These harsh conditions have led to widespread forest die-off, resulting in significant and escalating losses. Therefore, conducting physiological and molecular mechanism studies on the response of *P*. *massoniana* to low-temperature stress will provide theoretical support for the breeding of new cold-resistant varieties of *P*. *massoniana*. However, there are currently few investigations on *P. massoniana*’s cold tolerance. Only our team has carried out some research in the early stages. Results showed that during the cooling process, *P. massoniana* regulates the trend of endogenous protective enzymes through changes in endogenous abscisic acid (ABA). It also coordinates the interaction of multiple enzymes to resist low temperature. Materials with strong cold tolerance have higher contents of chlorophyll, soluble sugars, and free proline. They also have lower contents of soluble proteins and malondialdehyde [21,22,23,24]; however, the materials used in these studies have significant limitations. Recent research predicts that future climate change will lead to further northward expansion of *P. massoniana* [25].

Evaluating cold-tolerant families of *P. massoniana*, screening high-quality cold-tolerant germplasm, analyzing the physiological and biochemical mechanisms of cold-tolerant families, and selecting and breeding cold-tolerant families are effective ways to improve *P. massoniana*’s cold-tolerance ability, cope with climate change, and ensure *P. massoniana*’s healthy development. This paper uses materials from the cold-tolerant and non-cold-tolerant families to investigate the response mechanism of low-temperature stress on the aboveground tissue morphology, physiological indexes, microstructure, and chlorophyll fluorescence characteristics of *P. massoniana*, in order to provide a reference for cold-tolerant families selection and mechanism research of *P. massoniana*.

## 2. Results

### 2.1. Morphological Characteristics of Different Cold-Tolerant Families under Low-Temperature Stress

After 24 h of stress at 10 °C and 10 °C R, the morphology of different cold-tolerant families (cold-tolerant, 20-628 and cold-sensitive, 20-654) did not change obviously after recovery to 25 °C. The needles and leaves of the two families showed slight wilting at 0 °C. They were able to return to normal growth after recovery to 25 °C. Both stems and leaves of the two families showed water loss, wilting, and frost damage at −5 °C. The cold-tolerant family’s needles wilted less severely, and the needles and terminal buds remained semi-erect (Figure 1).

### 2.2. Effects of Low-Temperature Stress on Chlorophyll Fluorescence Characteristics of Different Cold-Tolerant Families

The Fo of the two families tended to increase as the temperature decreased. Compared to the control at 25 °C, the minimum fluorescence value (*Fo*) increased significantly under 0 °C and −5 °C stress, reaching a highly significant level at −5 °C. After recovering at 25 °C for 24 h, the *Fo* values of the two families under 0 °C stress began to decrease but remained significantly higher than those observed under the 25 °C and 10 °C treatments (Figure 2). The maximum photochemical efficiency (*Fv*/*Fm*) of the two families was not significantly different from that of the control under the 10 °C treatment. However, there was a significant difference between the treatment at 10 °C R and the control. The *Fv/Fm* was significantly lower than that of the control under 0 °C and −5 °C stress. After the 0 °C R treatment, the *Fv/Fm* of the two families increased. In contrast, the *Fv/Fm* decreased sharply under −5 °C stress (Figure 2). Actual photochemical efficiency (*ΦPSII*), electron transport rate (ETR), and photochemical quenching (*qP*) indexes were significantly lower than that of the control under 10 °C, 0 °C, and −5 °C stress and showed a decreasing trend with decreasing temperature. After 24 h of recovery at 10 °C R, the *ΦPSII*, *ETR*, and *qP* of the two families increased to different degrees. The *ΦPSII*, *ETR*, and *NPQ* indexes of the cold-tolerant family were significantly higher than those of the cold-intolerant family. Additionally, *qP* was higher in the cold-tolerant family compared to the cold-intolerant family, but this difference was not significant. This indicates that at a low temperature of 10 °C, the cold-resistant family has a strong recovery ability (Figure 2).

### 2.3. Effects of Low-Temperature Stress on the Physiology of Families of Different Cold-Tolerance

#### 2.3.1. Membrane Permeability

The relative conductivity of different tissues in seedlings of different families of *P. massoniana* under low-temperature stress showed the same trend, which increased with the decrease of temperature, and decreased after 10 °C R and 0 °C R treatments relative to the 10 °C and 0 °C treatments. The differences were that the conductivity of the non-tolerant family was significantly reduced after 0 °C R treatment in stems relative to the 0 °C treatment, while the difference between the tolerant families was not significant. In the needles and leaves, the two families showed basically the same performance, and reached a significant difference with the 0 °C treatment, which indicated that the tolerant family was able to recover earlier from the damage of the stem cell membranes caused by the low temperatures, and the trend of the change of MDA content was similar to that of the conductivity (Table 1 and Table 2).

#### 2.3.2. Protective Enzymes

The trends of changes in the stems and needles of the two families under low-temperature stress were different, showing a decreasing–ascending–decreasing trend in the stems, while in the needles the overall trend was on the rise, and the 10 °C R and 0 °C R treatments relatively increased the SOD activity in stems. The SOD activity was significantly reduced after 10 °C R in needles, and the SOD activity of the cold-intolerant family rose faster in 0 °C R treatment. The SOD activity in stems was significantly lower than that of the control under −5 °C stress, whereas the SOD activity in needles was in the ascending stage at any one time (Table 1 and Table 2). Different low temperatures and low-temperature recovery treatments had little effect on the POD activity in stems. The pattern of change in the POD activity in needles and leaves of the two families was consistent. Both showed reduced activity relative to the control at 10 °C, while the other treatments were in the ascending stage. Additionally, the SOD and POD levels of the cold-tolerant family were higher than those of the non-cold-tolerant family at 25 °C (Table 1 and Table 2).

#### 2.3.3. Osmoregulatory Substances

Proline content in stems and needles of the two families showed an increasing trend despite the temperature reduction, with the highest accumulation at 0 °C. However, in needles, the proline content at 10 °C R was higher than that under 10 °C stress. For other recovery treatments, the proline content showed a decreasing trend relative to the same temperature. The changes in soluble sugar content in the different tissues of the two families under low-temperature stress varied. In the cold-tolerant family, the soluble sugar content in the needles showed an increasing trend despite the temperature reduction. In contrast, the cold-intolerant family exhibited a decline in soluble sugar content after reaching the highest value during the 10 °C treatment, although it remained higher than the control. Additionally, the 0 °C R treatment significantly increased the soluble sugar content of the cold-intolerant family. In stems, the opposite trend was observed, except for the cold-tolerant family, the content of which increased in stress at −5 °C, and decreased in other treatments. The 0 °C R treatment significantly increased the soluble sugar content of two families, but 10 °C R was a decreasing trend. Proline and soluble sugar contents in stems were lower in the cold-tolerant family than in the non-cold-tolerant family at 25 °C (Table 1 and Table 2). The soluble protein contents of the two families were in an increasing trend in stems under low-temperature stress, except for the cold-intolerant family which decreased in the 0 °C treatment, while in needles it showed an increasing–decreasing–increasing trend. The 0 °C R treatment increased the soluble protein contents in needles, and the soluble protein contents of the cold-tolerant family increased significantly more than that of the cold-intolerant family at −5 °C stress (Table 1 and Table 2).

### 2.4. Effects of Low-Temperature Stress on the Anatomical Structure of Different Cold-Tolerant Families

The cross-sections of the stems of *P. massoniana* seedlings were epidermis, cortex and xylem from outside to inside. During normal growth, the epidermal and cortical cells of the stems of the two families were neatly arranged and tightly packed, and the structure of the stems did not undergo obvious changes under the treatments of 10 °C and 10 °C R. The morphological changes of the cells in the cross-sections under the treatment of 0 °C were not obvious, but the thin-walled cells of part of the cortex in the longitudinal section were obviously deformed with the cell gaps increasing, and the cells in the cortex under the treatment of −5 °C were crumpled with the loss of water, and some epidermis was detached from the cortex. In the −5 °C treatment, the epidermal cells lost water and crumpled, part of the epidermis was detached from the cortex, some thin-walled cells of the cortex were ruptured and fused, and the arrangement of the cells was disordered (Figure 3). The cross-sections of the needles of the *P. massoniana* seedlings were completed in order from outside to inside: complete epidermis, phloem cells, endothelium, thin-walled tissue, and xylem. There were no obvious changes in cell morphology under 10 °C and 10 °C R treatment. The 0 °C and 0 °C R treatment showed no obvious changes in other tissue cells except for the deformation of some chloroplasts. The structure of some epidermal cells was damaged in the −5 °C treatment of needles, and the leaf pulp cells were ruptured and deformed. The structural integrity of the ring-shaped leaf pulp cells of the cold-intolerant family was destroyed, and the cells of the endodermis and thin-walled tissue were deformed severely (Figure 4).

### 2.5. Effects of Low-Temperature Stress on Starch Grain Accumulation in Different Cold-Tolerant Families

Starch granules were scattered and lightly colored in the thin-walled tissue of the stem cortex under 25 °C, 10 °C, and 10 °C R treatments. Under 0 °C stress, the number of starch granules in the cortical cells of the stems increased, and the granules became larger. The cold-tolerant family showed significantly more starch accumulation compared to the less cold-tolerant family. No obvious starch granules were found in the stems at −5 °C (Figure 3). Under normal growth conditions, the cold-tolerant family displayed a large number of starch granules in their acinar cells. These granules were lighter in color compared to the control. Upon the 10 °C treatments, there was a significant decrease in the number of starch granules; however, their number sharply increased with the 0 °C treatment. Overall, the cold-tolerant family had more starch granules than the non-cold-tolerant family. Both the number and color of the granules were reduced under the 10 °C and 0 °C treatments. No starch granules were detected in the acinar cells of either family following the −5 °C treatment (Figure 4).

### 2.6. Effect of Low-Temperature Stress on Lignin Content of Different Cold-Tolerant Families

Under 10 °C treatment, the xylem of the stems of the two families was darker in hydrochloric acid–mesoterphenol solution than in that of the control, and with the decrease in the stress temperature, in addition to the deepening of the xylem coloring, the epidermal cell walls of the stems were also stained red (Figure 5). In conifers, lignin was mainly found in epidermal cell walls and xylem. No significant changes occurred in the two families at 10 °C compared with the control. After treatment at 0 °C, the epidermal cell walls were slightly darker. Following treatment at −5 °C, the epidermal cell walls of the cold-tolerant family were obviously reddened. Additionally, some reddening of the cell walls was observed in the thin-walled tissues of the xylem (Figure 6).

## 3. Discussion

*P. massoniana* is mainly distributed south of Qinling Mountains-Huaihe River of China, and is susceptible to low temperatures during its spread to the north. By assessing the performance of germplasm resources in low-temperature environments, we can identify and utilize potential cold-resistant germplasm resources, providing valuable genetic resources for future breeding efforts [26]. Morphological performance is the most intuitive manifestation under low-temperature stress, especially in above-ground tissues such as leaves and stems. Low-temperature-tolerant materials can be evaluated by performance indexes, especially by using seedlings, which are the most susceptible to low-temperature damage. In this study, both stems and leaves of the two families showed water loss wilting and frost damage under −5 °C stress. However, the cold-resistant family exhibited a more cold-tolerant phenotype at low temperatures of 10 °C and above. The needles of the cold-tolerant family showed less wilting, and the needles and terminal buds remained in a semi-erect state. Considering fluorescence parameters, cellular structure, and lignin content in the present study, the cold-tolerant family had more parenchyma cells, lignin, and starch content, which enhanced photosynthesis and strengthened the protective enzyme system capacity. Whether the upper epidermal thickness, spongy tissue thickness, tissue compactness, and stomatal density have an effect needs further study [27,28].

Photosynthesis is one of the most sensitive physiological and biochemical processes to low-temperature stress [29,30]. In previous studies, all coniferous species showed a decrease in *Fv/Fm* after low-temperature stress, but the decrease was slower in cold-tolerant materials [31]. In the present study, we found that the *Fv/Fm*, *ΦPSII*, *ETR*, and *qP* of the two families showed an overall decreasing trend with decreasing temperature, while *Fo* and *NPQ* showed an increasing trend, which is in agreement with the results of the previous studies on *Pinus densiflora* [32], *Pinus densata* [33], and other plants, suggesting that the PSII reaction center of *P. massoniana* seedlings was damaged. Recent studies have shown that chlorophyll fluorescence (ChlF) analysis, with its advantages of being rapid, easy to operate, low-cost, and highly sensitive, can be used to detect the effects of various stresses on plants. Among the chlorophyll fluorescence parameters, the photochemical quenching coefficient (*qP*) in the PSII reaction center can be used for early stress detection and is the most accurate indicator for screening the effects of environmental stress on plants [34]. This study found that under 0 °C treatment, the *qP* of the cold-resistant family was significantly higher than that of the non-cold-resistant family. This difference in *qP* may explain the different mechanisms of the PSII system in different cold-resistant families, potentially becoming the most effective method for the rapid identification of tree species.

At the same time, the Fo values of the two families reached the highest and the other index values reached the lowest values at −5 °C, indicating that the PSII reaction center was reversibly inactivated, the photosynthetic structure was damaged, and photosynthesis was inhibited in the ponytail pine seedlings under low-temperature stress below 0 °C. *NPQ*, on the other hand, increased under low-temperature stress above 0 °C, suggesting that the plant itself has a certain ability to repair and alleviate the damage caused by low temperature at this time [13,14]. Recent studies have shown that under relevant stresses, the inhibition of photosynthesis-related proteins leads to an enhanced ROS-scavenging capacity during oxidative stress, which is the main reason for the cold tolerance of resistant varieties [35].

In order to adapt to or resist the low temperature environment, a series of physiological changes occurred in the plants. In this study, the conductivity of 0 °C R treatment of cold-tolerant family was still engaged against 0 °C treatment, while that of the cold-intolerant family decreased, indicating that the 0 °C low temperature had caused irreversible damage to the cold-intolerant family [36]. Meanwhile, low-temperature stress causes plants to activate antioxidant enzyme systems to scavenge excessive reactive oxygen species, thus maintaining cell membrane stability [10]. In this study, we showed that SOD and POD in the continuous low-temperature stress and same temperature recovery treatments generally showed an increase, while in the 0 °C R treatment of the cold-intolerant family, SOD activity rose faster, indicating that the cold-intolerant family would produce more reactive oxygen species at −5 °C in the needles and leaves. The SOD and POD were still rising, indicating that the protective enzyme system still had effects under the current treatment and could be found in the stems and leaves. At −5 °C, SOD and POD in the needles and leaves were still increasing, indicating that the protective enzyme system had not completely “broken defense”, and it could be found that SOD in stems treated with −5 °C had been significantly reduced while POD was still increasing, suggesting that SOD was mainly involved in the early stage of stress and POD played a protective role in the late stage of stress [8]. Soluble sugars and soluble proteins have obvious effects on improving cold tolerance of plants [37], while soluble sugars in the needles and leaves and soluble proteins in the stems of the non-cold-tolerant family showed a decreasing trend at 10 °C and 0 °C, respectively, which also suggests that the more cold-tolerant materials have higher contents of soluble sugars and soluble proteins under low-temperature stress, which shows stronger antioxidant and cold-tolerance abilities [38,39]. In this study, we expressed that the recovery from low-temperature treatments at 10 °C and 0 °C changed the physiological and biochemical indexes in plants, which could play the role of low-temperature exercises and improve cold tolerance [36].

A simple and rapid evaluation system by measuring physiological and biochemical indicators of different cold-tolerant germplasm resources and combining them with phenotypic traits is the main evaluation method for the plants [40]. Under low-temperature stress, the physiological indicators of different cold-resistant varieties show significant changes. The mechanisms for coping with low-temperature stress mainly include the following: firstly, increasing intracellular osmotic potential through the synthesis of osmotic regulatory substances, and secondly, scavenging reactive oxygen species (ROS) through antioxidant enzymes. When evaluating cold resistance, the performance of different physiological indicators varies under low-temperature stress. The cold resistance response mechanism of plants is relatively complex. Relying on a single indicator to determine cold resistance is overly simplistic. Therefore, establishing the cold resistance level of *P. massoniana* germplasm resources through multiple indicators makes the evaluation more scientific. Moreover, since *P. massoniana* is distributed in different climatic zones and ecological types in China, using simple physiological indicators to analyze its evolutionary mechanisms for resisting low temperatures is another approach. This approach also helps in understanding how *P. massoniana* adapts to climate change on a large scale and maintains dominant communities in natural forests [41].

Plant tissue structure has been shown to correlate with plant cold tolerance [42]. In this study, we found that the two families in general did not have significant changes in stem and needle tissue structure under 0 °C and 0 °C R treatments, or the plants could undergo self-recovery, and the indicators of the cold-tolerant family were due to the non-cold-tolerant family, whereas the collapse and rupture of the cellular morphology, cellular gaps, and inter-cellular arrangement under −5 °C indicated that freezing below 0 °C caused unlikely damage to the seedlings of *P. massoniana*.

When plants are subjected to low-temperature stress, nutrients like sugars in the form of starch granules are also accumulated in plants to improve their cold tolerance [43,44]. In this study, we found that the content of starch granules in needles and stems of the cold-tolerant family increased under 10 °C and 0 °C stress, indicating that low temperature promoted the synthesis of starch granules in cold-tolerant plants, which was similar to the result of increasing starch granule accumulation in tissue volume before overwintering in peonies in the Central Plains [45]. Lignin is one of the end products of the phenylpropane metabolic pathway, which plays an important role in improving plant stress tolerance [17,18,46]. In this study, we found that lignin in the stems and needles of seedlings under low-temperature stress accumulated significantly with the decrease of temperature, and the −5 °C treatment of the cold-tolerant family accumulated more lignin, suggesting that the cold-tolerant family is able to enhance its own cold-resistance by increasing the content of lignin, which has also been confirmed in studies of different cold-tolerant poplar varieties and different cold-tolerant conifers in the north and south of the China [4]. Therefore, starch acts as a protective agent against abiotic stress, playing a defensive role. Studies have analyzed how starch alters sugar accumulation under abiotic stress and alleviates excessive sugar accumulation in both photosynthetic and non-photosynthetic tissues. Our current research, along with previous studies, indicates that starch plays a crucial role in the response of *P. massoniana* to abiotic stress [19,20]. The differences in starch accumulation under low-temperature stress among different cold-resistant families will provide valuable insights into the mechanisms of cold resistance.

## 4. Materials and Methods

### 4.1. Plant Materials and Environmental Controls

The cold-tolerant (20-628) and cold-sensitive (20-654) families chosen for this study were selected from 321 *P. massoniana* families evaluated in the preliminary stage of the investigation by repeated natural low-temperature stress [23]. After the cotyledons had emerged from the shells of the seedlings, the full seeds of the two families were selected, sterilized with 0.1% potassium permanganate, germinated (by immersing the seeds in warm water at 45 °C for 24 h), drenched dry, uniformly sown in nursery beds for cultivation, and then transferred into pots (12:18 cm wide:18 cm high) filled with yellow soil. For 30 days, the plants were placed in an artificial climate chamber (temperature: 27 °C, humidity: 60%, day/night photoperiod: 16 h/8 h, light intensity: 1200 lx). During this period, the plants were managed according to the technical regulations for *P. massoniana* seedling cultivation [47]. The plants were sprayed with calcium superphosphate fertilizer, which has an effective phosphorus content of 1/1000. The fertilizer was sprayed onto the needles until dripping, and the substrate cups showed water seepage. This process was repeated once every 10 days.

Six treatment groups were set up for different cold-tolerant families, including 25 °C (CK), 10 °C, 0 °C (chilling injury), −5 °C (freezing injury), 10 °C R, and 0 °C R, with three replications of three pots in each group. Before treatment, the cultivated 30-day-old seedlings were transferred to an artificial climate chamber for three days (temperature: 25 °C, humidity: 60%, day/night photoperiod: 16 h/8 h, light intensity: 1200 lx), during which they were routinely maintained at a temperature of 25 °C and under the same conditions. The cooling procedure was carried out at 4 °C/h with 25 °C as the control (CK), 10 °C, 0 °C, and −5 °C low-temperature stress treatment, and the stress was continued for 24 h, followed by collection after reaching each treatment time point. After reaching 10 °C and 0 °C stress levels, a portion of the material was placed in an artificial incubator for the response experiment. The warming process was carried out at 4 °C/h for rewarming, and sampling was performed after reaching 25 °C for 24 h, which was expressed as 10 °C R and 0 °C R, respectively. Morphological observations and photographs were taken at the end of each treatment, while chlorophyll fluorescence was measured, and stem and needle samples were measured or preserved according to the requirements of each index, and three replications were carried out for each sample.

### 4.2. Experimental Measurements

#### 4.2.1. Chlorophyll Fluorescence Parameters Determination

The FMS-2 portable pulse-modulated fluorometer (Hansatech, Norfolk, UK) was used. According the to methodology of specification, the needles were dark-adapted for 40 min before the determination. Afterwards, the indexes of *Fo* after dark adaptation were examined according to the requirements, the maximum fluorescence (*Fm*), *Fv/Fm*. When the fluorescence quickly fell from *Fm* to *Fo* (5 s), steady-state fluorescence (*F*s) was measured with continuous actinic light. After superimposing a saturating pulse, the maximum fluorescence value (*Fm*′) was measured after the plant adapted to a specific light intensity, and a saturating pulse was applied. Following the closure of the actinic light for 3 s of dark adaptation, after photosynthesis reached a steady state, the fluorescence value under far-red light (*Fo*′) was measured. Subsequently, *ΦPSII* was determined by the instrument. The photochemical quenching coefficient (*qN*) was calculated using the formula *qN* = (*Fm* − *Fm*′)/(*Fm*′ − *Fo*′), and the non-photochemical quenching coefficient (*NPQ*) was calculated using the formula *NPQ* = (*Fm* − *Fm*′)/*Fm*′. The electron transport rate (ETR) was calculated using the formula ETR = *ΦPSII* × 0.85 × PAR × 0.5. Each treatment was measured in triplicate, with three needles measured per plant.

#### 4.2.2. Relative Conductivity

According to the method of Lu et al. [23], the relative conductivity was measured using a conductivity meter (DDS-11A, Leici, Shanghai, China). Samples were rinsed three times with ultrapure water and dried with clean filter paper. Roots were cut into 0.5 cm segments using clean scissors and placed into 50 mL centrifuge tubes. Each treatment had three replicates. Subsequently, 25 mL of ultrapure water was added, and the samples were vacuumed at room temperature for 30 min, then left to stand for 20 min. The initial conductivity (*R*_1_) was measured at room temperature. The centrifuge tubes were then placed in a boiling water bath for 20 min and removed. The boiling conductivity (*R*_2_) was measured after the temperature was reduced to room temperature. Relative conductivity = *R*_1_/*R*_2_ × 100%.

#### 4.2.3. Physiological Index Measurement

The activities and contents of superoxide dismutase (SOD), proline (Pro), malondialdehyde, soluble sugars, and soluble proteins were determined using Suzhou Keming Bio-technology Co. Ltd. (Suzhou, China) kits, and absorbance was measured using Shanghai Shanpu Biotech Co. Ltd.’s (Shanghai, China) SuPerMax 3100 enzyme labeling instrument [23].

#### 4.2.4. Microscopic Observation

The stems and leaves were cut from the middle part by scalpel. Each segment was 0.5 cm long. The stems and leaf tissues of the treated seedlings were fixed in FAA solution (90 mL alcohol, 5 mL formalin, 5 mL glacial acetic acid). After placing the tissues in FAA fixative solution, they were placed in a vacuum environment for 15 min to remove air from the tissue cells, allowing the fixative solution to penetrate the cells more effectively. The fixative solution was replaced promptly after vacuuming. According to the method of Lu et al. [23], the tissues underwent dehydration, clearing, paraffin infiltration, embedding, sectioning, mounting, dewaxing, clearing, rehydration, and staining. During the paraffin infiltration step, paraffin with a melting point of 54 °C was used at 42 °C for 2 days, followed by replacement with paraffin with a melting point of 58 °C and continued infiltration at 60 °C for 4 h. The thickness of the sections was 10 μm. The sections were stained with solid green-iodine potassium iodide staining. The sections were placed in iodine-potassium iodide solution for 10 min, rinsed twice with distilled water, then placed in 95% alcohol for 2 min, followed by staining with 1% solid green dye solution for 1 min and 30 s. The sections were decolorized through a series of absolute alcohol steps. The sections were sealed with neutral resin and placed under the microscope to observe their morphology. The embedding machine was the EG 1150H (LEICA, Wetzlar, Germany). The automatic microtome was the RM2235 (LEICA, Wetzlar, Germany). The microscope was the DS-Ri2 (Nikon, Minato, Japan).

#### 4.2.5. Observation of Lignin Content

In the early stage, the same process as that of microscopic observation of the preparations was carried out with resorcinol staining method. We added a drop of concentrated hydrochloric acid to the slides containing the tissues and let them sit for 5 min to acidify the cells. Then, we added a drop of phloroglucinol solution and let them sit for 4 min. After that, we covered them with a coverslip and observed the changes in lignin content in the tissues under a microscope. The embedding machine was EG 1150H (LEICA, Wetzlar, German), the automatic slicing machine was RM2235 (LEICA, Wetzlar, German), and the microscope was DS-Ri2 (Nikon, Minato, Japan).

### 4.3. Data Analysis

All data were processed using Excel, analyzed via two-way analysis of variance (ANOVA) using SPSS 13.0 software, and plotted using GraphPad Prism 7.0. The normality distribution was assessed using the Kolmogorov–Smirnov test (K-S test) in SPSS. If the *p*-value is greater than the set significance level (0.05), the data are considered to follow a normal distribution.

## 5. Conclusions

The cold-tolerant and non-cold-tolerant families of *P*. *massoniana* exhibited similar trends in phenotypic morphology, physiology, biochemistry, and tissue structure changes under low-temperature stress. They responded to the stress by increasing *ΦPSII* and ETR indexes, enhancing membrane permeability, improving the activity of protective enzymes, boosting osmotic regulating substances, and increasing the levels of starch grains and lignin content in the xylem. The cold-tolerant family accumulated more nutrients and metabolites, thereby demonstrating improved cold tolerance. A temperature of 0 °C was the critical value for the response to low temperature of the experimental materials, and the moderate low-temperature exercise at 10 °C or 0 °C was able to improve the ability of the seedlings to withstand low temperatures. Further molecular biology techniques and field experiments are needed to further analyze and verify the deeper mechanism. The results of this study enhance our understanding of low-temperature adaptability in conifer species, represented by *P*. *massoniana*. This will provide valuable new insights and directions for breeding and research on the mechanisms of cold tolerance in coniferous trees.

## Figures and Tables

**Figure 1 plants-13-02229-f001:**
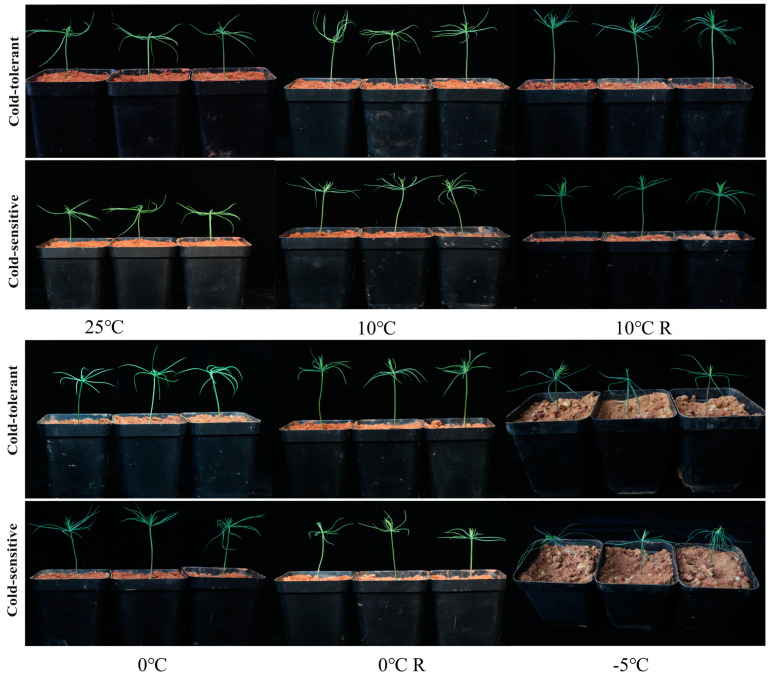
Morphological characteristics of *P. massoniana* under different low-temperature stresses. We conducted a study on the morphological characteristics of *P. massoniana* seedlings under treatments of 25 °C (CK), 10 °C, 10 °C recovery, 0 °C, 0 °C recovery, and −5 °C. “20-628” represents the cold-tolerant family, while “20-654” represents the cold-sensitive family.

**Figure 2 plants-13-02229-f002:**
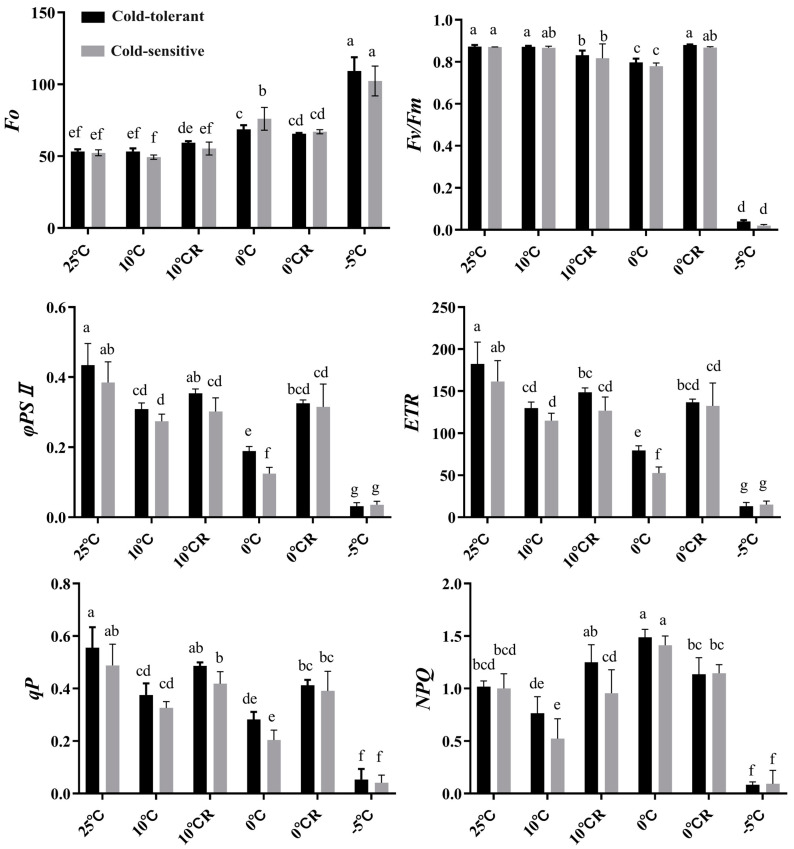
Effects of low-temperature stress on chlorophyll fluorescence characteristics of families of different cold-resistance. Different lowercase letters indicate significant differences between different families under the same temperature (*p* < 0.05).

**Figure 3 plants-13-02229-f003:**
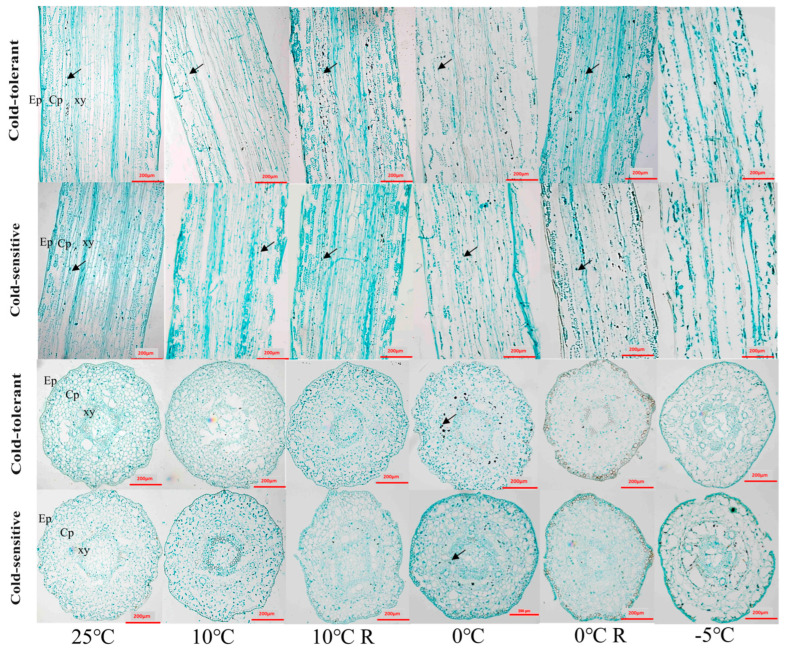
Effects of low-temperature stress on stem anatomical structure of *P. massoniana* seedlings. Ep is epidermis, Cp is cortex, Xy is xylem. The black arrow shows starch granules.

**Figure 4 plants-13-02229-f004:**
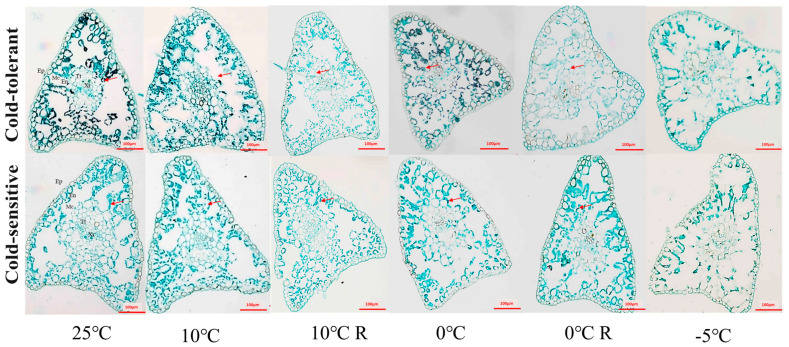
Effect of low-temperature stress on anatomical structure of *P. massoniana* seedling needles. Ep is epidermis, Mc is mesophyll cell, En is endodermis, Tt is parenchymatous tissue, and Xy is xylem; the red arrow shows starch granules.

**Figure 5 plants-13-02229-f005:**
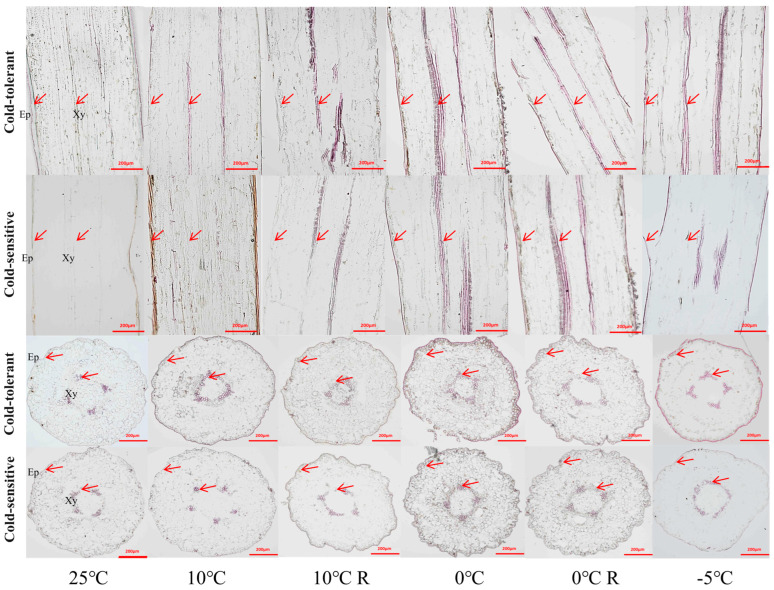
Effects of low-temperature stress on lignin content in stem of *P. massoniana* seedlings. Ep is epidermis, Xy is xylem, and the red arrow shows lignin.

**Figure 6 plants-13-02229-f006:**
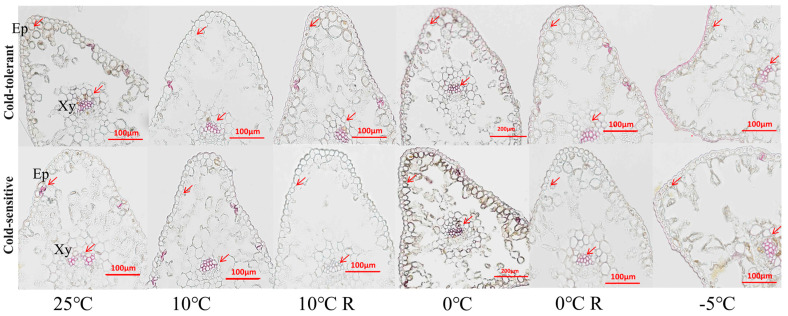
Effects of low-temperature stress on lignin content in leaves of *P. massoniana* seedlings. Ep is epidermis, Xy is xylem, and the red arrow shows lignin.

**Table 1 plants-13-02229-t001:** Effects of low-temperature stress on physiological indexes of stem in different cold-resistant families.

Family	Temperature/°C	Relative Conductivity (%)	MDA Content (nmoL/g)	Proline Content (μg/g)	Soluble Sugar Content (μg/g)	Soluble Protein (μg/g)	SOD Activity (U·g^−1^·FW)	POD Activity (μ·g^−1^·min^−1^)
cold-tolerant	25	0.31 ± 0.02 bcd	1.08 ± 0.04 e	85.35 ± 9.21 de	6.54 ± 0.37 ab	10.95 ± 0.56 e	182.02 ± 10.23 cd	2954.60 ± 92.36 ab
	10	0.28 ± 0.03 cde	1.38 ± 0.09 de	94.66 ± 12.00 cde	5.73 ± 0.60 bc	21.01 ± 2.38 abc	146.70 ± 14.06 e	2741.71 ± 161.62 ab
	10 R	0.28 ± 0.03 de	1.21 ± 0.05 e	86.98 ± 5.37 de	2.78 ± 0.27 e	22.34 ± 1.18 ab	158.60 ± 33.45 de	2815.33 ± 199.75 ab
	0	0.32 ± 0.01 bc	1.90 ± 0.40 c	128.93 ± 4.01 a	4.08 ± 0.42 d	22.48 ± 2.68 ab	187.70 ± 17.08 bcd	3573.67 ± 272.82 a
	0 R	0.26 ± 0.03 e	1.62 ± 0.16 cd	111.08 ± 9.03 abc	7.25 ± 0.69 a	21.14 ± 4.12 abc	212.51 ± 27.31 abc	2878.43 ± 199.67 ab
	−5	0.95 ± 0.01 a	3.53 ± 0.06 a	98.70 ± 18.60 cd	7.03 ± 0.45 ab	23.32 ± 0.49 a	83.56 ± 6.07 f	2916.91 ± 181.75 b
cold-sensitive	25	0.30 ± 0.01 bcd	1.15 ± 0.23 e	95.19 ± 6.67 cde	7.29 ± 1.37 a	11.62 ± 0.80 e	218.15 ± 22.64 ab	2747.54 ± 134.85 ab
	10	0.31 ± 0.01 bcd	1.32 ± 0.11 de	75.65 ± 6.09 e	7.51 ± 0.31 a	18.94 ± 1.24 abc	143.52 ± 15.54 e	2366.18 ± 134.72 b
	10 R	0.29 ± 0.04 cde	1.24 ± 0.16 e	88.11 ± 11.66 de	4.68 ± 0.85 cd	18.01 ± 2.17 cd	161.81 ± 23.18 de	2745.23 ± 153.41 ab
	0	0.34 ± 0.00 b	2.30 ± 0.24 b	121.14 ± 16.22 ab	2.61 ± 0.03 e	15.44 ± 1.24 d	185.52 ± 17.18 bcd	2798.71 ± 119.31 ab
	0 R	0.28 ± 0.02 de	1.63 ± 0.27 cd	91.66 ± 11.41 cde	7.75 ± 1.61 a	21.46 ± 2.18 abc	222.00 ± 18.54 a	3432.68 ± 235.71 ab
	−5	0.96 ± 0.02 a	3.58 ± 0.24 a	101.39 ± 15.4 bcd	2.37 ± 0.3 e	23.19 ± 1.43 a	86.56 ± 2.56 f	2394.50 ± 165.32 ab

Note: MDA represents malondialdehyde; SOD represents superoxide dismutase; POD represents peroxidase. No the same lowercase letters among different temperatures in the same family indicate significant differences (*p* < 0.05).

**Table 2 plants-13-02229-t002:** Effects of low-temperature stress on physiological indexes of leaves in different cold-resistant families.

Family	Temperature/°C	Relative Conductivity (%)	MDA Content (nmoL/g)	Proline Content (μg/g)	Soluble Sugar Content (μg/g)	Soluble Protein (μg/g)	SOD Activity (U·g^−1^·FW)	POD Activity (μ·g^−1^·min^−1^)
cold-tolerant	25	0.25 ± 0.00 e	1.08 ± 0.02 ef	76.13 ± 2.25 d	9.67 ± 0.58 cde	19.01 ± 1.53 ef	129.33 ± 6.03 c	1509.51 ± 96.47 cd
	10	0.3 ± 0.01 d	1.27 ± 0.35 def	81 ± 3.2 d	12.91 ± 0.73 bc	26.57 ± 1.76 bc	188.87 ± 30.63 ab	1147.95 ± 135.49 de
	10 R	0.25 ± 0.02 e	1.24 ± 0.26 def	117.24 ± 9.77 c	11.76 ± 1.44 bcd	25.29 ± 2.8 cd	136.87 ± 1.96 c	1550.05 ± 136.4 cd
	0	0.35 ± 0.02 c	2.19 ± 0.2 c	154.66 ± 7.65 a	14.19 ± 1.55 ab	14.76 ± 1.8 f	173.15 ± 9.2 bc	1660.59 ± 123.39 bc
	0 R	0.24 ± 0.03 ef	1.48 ± 0.14 de	88.34 ± 13.58 d	14.65 ± 1.09 ab	24.39 ± 1.43 de	202.6 ± 30.76 ab	2227.76 ± 39.42 a
	−5	0.97 ± 0.02 a	2.58 ± 0.22 ab	146.98 ± 10.09 ab	13.00 ± 3.71 bc	33.99 ± 3.17 a	204.31 ± 26.02 ab	2418.48 ± 97.34 a
cold-sensitive	25	0.23 ± 0.00 f	1.36 ± 0.30 def	104.75 ± 13.8 c	9.00 ± 1.00 de	19.00 ± 1.39 ef	125.33 ± 4.16 c	1044.14 ± 123.82 e
	10	0.29 ± 0.01 d	1.42 ± 0.24 def	54.75 ± 4.09 e	12.53 ± 0.87 bc	25.51 ± 3.65 cd	224.89 ± 34.3 ab	1081.06 ± 113.56 e
	10 R	0.24 ± 0.01 ef	1.06 ± 0.02 f	82.45 ± 4.27 d	7.62 ± 0.23 e	18.48 ± 0.33 ef	78.95 ± 7.71 d	1792.02 ± 93.18 bc
	0	0.39 ± 0.01 b	2.33 ± 0.31 bc	139.35 ± 12.89 ab	10.06 ± 1.39 cde	15.36 ± 3.48 f	218.31 ± 19.2 ab	1655.94 ± 90.04 bc
	0 R	0.24 ± 0.03 ef	1.57 ± 0.09 d	76.44 ± 4.86 d	16.53 ± 2.97 a	21.32 ± 1.14 de	219.6 ± 39.98 ab	2246.97 ± 106.82 a
	−5	0.98 ± 0.01 a	2.94 ± 0.11 a	136.43 ± 13.7 b	10.07 ± 2.27 cde	30.05 ± 3.92 ab	239.55 ± 30.26 a	2062.39 ± 60.63 ab

Note: MDA represents malondialdehyde; SOD represents superoxide dismutase; POD represents peroxidase. No the same lowercase letters among different temperatures in the same family indicate significant differences (*p* < 0.05).

## Data Availability

Data are contained within the article.

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
