# Peer review of "Effects of Low-Temperature Stress on Physiological Characteristics and Microstructure of Stems and Leaves of Pinus massoniana L."

_plants, 2024, doi:10.3390/plants13162229_

Round 1
Reviewer 1 Report
Comments and Suggestions for Authors
The manuscript titled 'Effects of Low Temperature Stress on Physiological Characteristics and Microstructure of Stems and Leaves of Pinus massoniana L.' presents valuable research on the morphological, physiological, and microstructural responses of P. massoniana to low temperatures. The results are particularly relevant for understanding the cold tolerance of this species and its potential adaptation strategies in the face of climate change. The manuscript is overall well-articulated and, if improved in the suggested areas, could significantly contribute to the field.
However, I have noticed numerous typos throughout the text. I strongly recommend a thorough proofreading and editing pass to correct these mistakes. Ensuring accuracy in spelling and grammar will enhance the manuscript's readability and credibility, which is essential for journal submission.
Furthermore, I recommend a careful review of the experimental data, particularly in the way significant differences in physiological responses of different families at different temperatures are analyzed and discussed. Such a review can lead to more nuanced conclusions and a stronger, more compelling discussion section. This detailed analysis will also allow for a better understanding of the mechanistic underpinnings of cold tolerance in this species, which could have important implications for forestry and conservation practices.
By addressing these points, I believe the manuscript will be a robust addition to the scientific literature and better positioned for publication in a reputable journal.
Abstract:
Could you please clarify some brief details about your experiment in the abstract? Was it conducted in a greenhouse? Did you use seedlings, and if so, how old were they?
Line 17: Delete “As a result,”
Introduction:
The introduction did not specify what is considered "low temperature" for the context of the study on Pinus massoniana. It broadly mentions low temperature stress and its effects on plants but lacks specific temperature values or ranges that define what "low temperature" means in this ecological and physiological study. Including such specifics would strengthen the introduction by providing clearer parameters for the study's conditions, making the research more precise and accessible to the reader. Additionally, you mentioned chlorophyll fluorescence, lignin concentration, and their roles in cold tolerance but it was brief. Expanding on how these factors specifically contribute to cold tolerance in Pinus massoniana could make the introduction more informative.
Other minor comments:
Line 40-42: Change “plant deaths” to “plant mortality”?
Line 43: delete “all”
Line 44: add “extensively” after “been”
Line 44: change “, as have the selection and breeding of new cold-resistant varieties [3-4].” to “Additionally, the selection and breeding of new cold-resistant varieties have also been explored [3-4]."
Line 45-47: Add Reference for “Plant cytoplasmic membranes, antioxidant enzyme systems, osmoregulatory substances, endogenous hormones, and other stress responses are formed in response to low-temperature stress, and defense is carried out by a complex regulatory system.”
Line 60: Change “because of” to “due to”
Line 61: “occurrence of extreme events” to “increasing frequency of extreme weather events”.
Line 61: What is 'ponytail pine'? Is this a common name for Pinus massoniana? If so, please ensure this is mentioned earlier before using the common name. Additionally, if you choose to use the common name, please apply it consistently throughout the manuscript. I noticed that 'ponytail pine' is used only twice in the document.
Line 62: Change “area has” to “areas have” then add “experienced numerous incidents of rain, snow, freezing conditions, and other low-temperature disasters. These harsh conditions have led to widespread forest die-off, resulting in significant and escalating losses.”, instead of adding “resulting in a large number of forests withering and then dying, resulting in massive losses that are becoming increasingly severe.”
Line 65-71: The transition from discussing research findings to introducing future predictions feels a bit abrupt. Consider adding a transitional phrase or sentence to create a smoother flow between these ideas and maintain the overall clarity and coherence of your text.
Line 75, 80: Species scientific name should be Italic.
Line 80: Make space between “cold” and “family”
Results:
· It may be easier for readers if you put “Cold-tolerant” and “Cold-sensitive” to the figures instead of just putting 20-628 and 20-654.
· I think the difference between families is scarcely pointed out in the result section. In your study, the different responses between the selected families suggest some level of stress or damage difference in physiology between the families, which is an important insight into how these families respond differently to low temperatures. Based on the above considerations, please revise the Results section.
2.2. Effects of low-temperature stress on chlorophyll fluorescence properties of different cold-tolerant families
· The significant difference in 𝐹𝑜 values between the two families at 0°C (but not at -5°C) should be mentioned and need further explanation in the discussion section. Also, you can indicate the fact that though 𝐹𝑜 was significantly different between families at 0°C, Fv/Fm was not in the discussion section.
Other minor comments:
Line 82-84: Delete
Line 85-91: Since there is no description of how you evaluated morphology in the result section, the description here is not clear. How did you analyze them? No statistical analysis on morphology?
Line 92 (Figure 1): Require informative description for Figure 1.
Line 96: In the introduction, you mentioned “chlorophyll fluorescence characteristics”. It may be better to use either characteristics or properties. Using terminology consistently can help readers.
Line 98: Change “decrease of temperature” to “decrease in temperature”.
Line 100: Change “after recovering in” to “after recovering at”.
Line 102: Adjust "that of the 25℃ and 10℃ treatments" to "those observed under the 25℃ and 10℃ treatments".
Line 98-102: Please break up the sentence to improve readability. It's currently quite challenging to follow.
Line 103: Fv/Fm at 10CR was significantly different compared to the control.
Line 102-106: Please break up the sentence to improve readability. It's currently quite challenging to follow. Or rewrite It to make it easier to follow the contents.
Line 108: Delete “temperature decreased.”
Line 126: italic
Line 162-167, 169-172, 172-177, 224-229, 229-236, 241-246: Please break up the sentence to improve readability. It's currently quite challenging to follow. Or rewrite It to make it easier to follow the contents. Using too many “and” in one sentence is not recommended. See the example below:
**Example for breaking up Line 229-236
“Under normal growth conditions, the cold-tolerant families displayed a large number of starch granules in their acinar cells. These granules were lighter in color compared to the control. Upon the 10℃ treatments, there was a significant decrease in the number of starch granules; however, their number sharply increased with the 0℃ treatment. Overall, the cold-tolerant families had more starch granules than the non-cold-tolerant families. Both the number and color of the granules were reduced under the 10℃ and 0℃ treatments. No starch granules were detected in the acinar cells of either family following the -5℃ treatment (Figure 4)”
Line 200, 201: Italic
Discussion:
The discussion section could be strengthened by delving deeper into the physiological mechanisms underlying the observed photosynthetic changes under cold stress. A comparative analysis highlighting the differences between the cold-tolerant and cold-sensitive families would clarify how specific adaptations confer cold tolerance. Additionally, broadening the scope to include ecological implications and practical applications, such as implications for forestry management and plant breeding, would make the findings more relevant and impactful. Enriching the discussion in these ways would provide a clearer understanding of the study’s broader significance.
- The manuscript mentions general trends in response to cold temperatures but could benefit from a more detailed comparison between the cold-tolerant and cold-sensitive families. Specifically, how do these families differ not just in the magnitude of change in photosynthetic parameters but also in their physiological responses and recovery capabilities? Highlighting these differences could provide valuable insights into the mechanisms of cold tolerance.
- The discussion on the reversibility of the damage at -5°C and the capacity for repair at temperatures above 0°C is crucial. However, it might be useful to discuss the long-term impacts of such stress if it were to occur repeatedly or persist over a longer period. How resilient are these plants to recurrent or prolonged cold stress?
- Extending the discussion to include ecological or evolutionary perspectives could enrich the analysis. How do these physiological responses to cold stress influence the ecological distribution and evolutionary adaptations of P. massoniana? How might these traits affect the plant's competitive ability and survival in natural settings under climate change scenarios?
Materials and Methods:
The materials and Methods section is missing many important information. By reading this section we should be able to redo your experiment. So please add more detailed information.
Additionally, I understand that 25℃ is for Control but I could not understand why the -5 ℃ treatment does not have -5 ℃R. Can you add a reason for that?
4.1. Design and measurement of greenhouse experiments
While the section provides a broad overview of the experimental setup, it could benefit significantly from a more detailed description of the experimental design and the specific treatments applied. It appears that six treatment groups were established, but the description of how these groups differ and the specific parameters of each treatment group are not clearly articulated. In the section title, it says “greenhouse experiments” but I could not find when the seedlings were moved to the greenhouse. Additionally, the experimental design is not adequately described. This information is crucial for understanding the experimental framework and ensuring the reproducibility of the results.
4.2.1. Chlorophyll fluorescence parameters determination
I noticed several areas where typos and redundant phrases in the section. To enhance the readability and accuracy, I recommend a thorough review to correct these typos. Additionally, simplifying sentences to remove unnecessary repetition will strengthen this section. The manuscript mentions the use of the FMS-2 portable pulse-modulated fluorometer twice in a way that seems redundant. I recommend consolidating these mentions into a single, clear description of the equipment used and its specific application in the context of your study.
Light-adapted measurements were not written here. Add all the parameters you measured using FMS-2.
The method of calculating indices such as 𝐹𝑣, 𝐹𝑣/𝐹𝑚, ФPSII, and NPQ should be explicitly stated, including any formulas used. This detail is crucial for replicability and for ensuring that other researchers can accurately compare your results with findings from other studies.
Please confirm and describe the sampling strategy used (e.g., number of leaves measured per plant, number of plants per treatment group) to ascertain the statistical validity of the results. This information is vital for assessing the robustness of your study's conclusions.
4.2.4. Microscopic observation
In the methodology section, it's crucial to maintain clarity and organization to ensure that readers can follow the procedures effectively. While the description you provided covers the steps involved in the experimental process, the repeated use of "and" makes the sentence long and somewhat cumbersome. Consider breaking down each step into separate sentences or using bullet points to list the actions taken. This will improve readability and make it easier for readers to understand and replicate the methodology. Additionally, providing context or explaining the purpose of each action can enhance the overall understanding of the experimental procedures. For instance, it would be beneficial to provide additional information or context regarding the "solid green-iodine potassium iodide staining" mentioned in the methodology. This could include details about the purpose of this staining technique, how it is performed, and its significance in the experimental process. Providing such information not only enhances the completeness of your methodology section but also helps readers understand the specific techniques and procedures used in your study.
4.3. Data analysis
The current description of the data analysis primarily outlines the software tools utilized but does not detail the checks for fundamental statistical assumptions required for ANOVA. It is essential to verify that the data meet the assumptions of normality and homogeneity of variances before conducting ANOVA, as deviations from these assumptions can lead to incorrect conclusions. If you have already performed these tests, please include a mention of this in the methodology section, specifying which tests were used and any subsequent steps taken to address potential issues.
Other minor comments:
Line 338: Move (W: H) to after “high”
Line 340: What do you mean by normally?
Line 351: Please provide an explanation for morphological observation. It is difficult to visualize how did you evaluate it.
Line 358: Provide the location of the product. I assume this is the right one “(Hansatech, Norfolk, UK)” but please check.
Line 363: delete “The” (not only “The” but also the entire paragraph has issues. Please re-rite this paragraph.
Line 364 Descriptions for Fo’, Fm’, are not correct.
Line 365: You mean “(ФPSII)”?
Line 387: Which data was processed? If all data were, add “All data”.
Line 387: I believe you did two-way ANOVA if so, change “ANOVA” to “two-way analysis of variance (ANOVA)”
Conclusion:
Based on the changes you will make in the discussion, please revise the conclusion as well.
Other minor comments:
Line 390: Italic for scientific name
**Please check all the scientific names in the manuscript and use italics for all.
Comments on the Quality of English LanguageAs I mentioned in the comments, the result section must be revised.
Author Response
Response to Reviewer
Thank you very much for the comments. We have revised the manuscript carefully according to the reviewer’s comments.
Abstract:
Could you please clarify some brief details about your experiment in the abstract? Was it conducted in a greenhouse? Did you use seedlings, and if so, how old were they?
A: We agree with the reviewer's suggestions. We have added “30-day-old seedlings from several cold-tolerant families were used as materials to simulate the morphological traits of needles and stems, chlorophyll fluorescence characteristics in artificial climate chamber.
Line 17: Delete “As a result,”
A: Yes,we did.
Introduction:
The introduction did not specify what is considered "low temperature" for the context of the study on Pinus massoniana. It broadly mentions low temperature stress and its effects on plants but lacks specific temperature values or ranges that define what "low temperature" means in this ecological and physiological study. Including such specifics would strengthen the introduction by providing clearer parameters for the study's conditions, making the research more precise and accessible to the reader. Additionally, you mentioned chlorophyll fluorescence, lignin concentration, and their roles in cold tolerance but it was brief. Expanding on how these factors specifically contribute to cold tolerance in Pinus massoniana could make the introduction more informative.
A: We have added this content. That is “Low temperature stress can be divided into chilling injury (0-15℃) and freezing injury (<0℃) [3]”(line 41-42)
Other minor comments:
Line 40-42: Change “plant deaths” to “plant mortality”?
A: Yes, we changed to plant mortality. (line 41)
Line 43: delete “all”
A: Yes, we did.
Line 44: add “extensively” after “been”
A: Yes, we did. That is “and physiological metabolism in plants under low temperature have been extensively studied in recent years.” (line 44)
Line 44: change “, as have the selection and breeding of new cold-resistant varieties [3-4].” to “Additionally, the selection and breeding of new cold-resistant varieties have also been explored [3-4]."
A: Yes,we did. That is“Additionally, the selection and breeding of new cold-resistant varieties have also been explored [4-5]”(line 44-45)
Line 45-47: Add Reference for “Plant cytoplasmic membranes, antioxidant enzyme systems, osmoregulatory substances, endogenous hormones, and other stress responses are formed in response to low-temperature stress, and defense is carried out by a complex regulatory system.”
A: Yes, we have added the reference.(line 47)
Line 60: Change “because of” to “due to”
A: Yes, we did. (line 60)
Line 61: “occurrence of extreme events” to “increasing frequency of extreme weather events”.
A: Yes, we did. ( line 61)
Line 61: What is 'ponytail pine'? Is this a common name for Pinus massoniana? If so, please ensure this is mentioned earlier before using the common name. Additionally, if you choose to use the common name, please apply it consistently throughout the manuscript. I noticed that 'ponytail pine' is used only twice in the document.
A: Yes,we have changed to P. massoniana. (line 61)
Line 62: Change “area has” to “areas have” then add “experienced numerous incidents of rain, snow, freezing conditions, and other low-temperature disasters. These harsh conditions have led to widespread forest die-off, resulting in significant and escalating losses.”, instead of adding “resulting in a large number of forests withering and then dying, resulting in massive losses that are becoming increasingly severe.”
A: Yes, we have changed to “the P. massoniana production areas have been subjected to numerous rain, snow, freezing, and low-temperature disasters, experienced numerous incidents of rain, snow, freezing conditions, and other low-temperature disasters. These harsh conditions have led to widespread forest die-off, resulting in significant and escalating losses. ” (line 61-64)
Line 65-71: The transition from discussing research findings to introducing future predictions feels a bit abrupt. Consider adding a transitional phrase or sentence to create a smoother flow between these ideas and maintain the overall clarity and coherence of your text.
A: Yes, we have added “Therefore, conducting physiological and molecular mechanism studies on the response of P. massoniana to low-temperature stress will provide theoretical support for the breeding of new cold-resistant varieties of P. massoniana.” (line64-66)
Line 75, 80: Species scientific name should be Italic.
A: Yes, we did. (line 75,79)
Line 80: Make space between “cold” and “family”
A: Yes, we did.(line79
Results:
It may be easier for readers if you put “Cold-tolerant” and “Cold-sensitive” to the figures instead of just putting 20-628 and 20-654.
A: We have changed the figures.Using Cold-tolerant instead of 20-628 and Cold-sensitive instead of 20-654. (line 82-83)
I think the difference between families is scarcely pointed out in the result section. In your study, the different responses between the selected families suggest some level of stress or damage difference in physiology between the families, which is an important insight into how these families respond differently to low temperatures. Based on the above considerations, please revise the Results section.
A:Based on the feedback, we further emphasized the differences in responses to cold stress between the two families in the results section.
2.2. Effects of low-temperature stress on chlorophyll fluorescence properties of different cold-tolerant families
- The significant difference in Fo values between the two families at 0°C (but not at -5°C) should be mentioned and need further explanation in the discussion section. Also, you can indicate the fact that though Fo was significantly different between families at 0°C, Fv/Fm was not in the discussion section.
A: We have added the corresponding content in the discussion section.
“However, there was a significant difference between the treatment at 10℃R and the control.”(line 92-93)
“This indicates that at a low temperature of 10℃, the cold-resistant family has a strong recovery ability”(line 100)
In discussion section, the contents were in line197-203 and line 204-205
Other minor comments:
Line 82-84: Delete
A:Yes, we deleted
Line 85-91: Since there is no description of how you evaluated morphology in the result section, the description here is not clear. How did you analyze them? No statistical analysis on morphology?
A:We conducted direct observations for morphological characteristics, and we believe they are quite straightforward. In fact, we collected data on wilting degree during the experiment. If the experts deem it necessary, we can upload this as supplementary figures.
Line 92 (Figure 1): Require informative description for Figure 1.
A: We have added a description of the information in Figure 1. (line473-476)
Line 96: In the introduction, you mentioned “chlorophyll fluorescence characteristics”. It may be better to use either characteristics or properties. Using terminology consistently can help readers.
A: We have changed to “chlorophyll fluorescence characteristics”(line 87)
Line 98: Change “decrease of temperature” to “decrease in temperature”.
A:We have changed the sentence into“The Fo of the two families tended to increase as the temperature decreased”.(line88)
Line 100: Change “after recovering in” to “after recovering at”.
A: We have changed to “After recovering at 25℃ for 24 hours”.(line89-90)
Line 102: Adjust "that of the 25℃ and 10℃ treatments" to "those observed under the 25℃ and 10℃ treatments".
A: We have changed to”“After recovering at 25℃ for 24 hours, the Fo values of the two families under 0℃ stress began to decrease but remained significantly higher than those observed under the 25℃ and 10℃ treatments”(line90-91)
Line 98-102: Please break up the sentence to improve readability. It's currently quite challenging to follow.
A: We have changed to “The Fo of the two families tended to increase as the temperature decreased. Compared to the control at 25℃, the Fo increased significantly under 0℃ and -5℃ stress, reaching a highly significant level at -5℃. After recovering at 25℃ for 24 hours, the Fo values of the two families under 0℃ stress began to decrease but remained significantly higher than those observed under the 25℃ and 10℃ treatments (Figure 2)”(line 88-91)
Line 103: Fv/Fm at 10CR was significantly different compared to the control.
A: We delete 10CR.
Line 102-106: Please break up the sentence to improve readability. It's currently quite challenging to follow. Or rewrite It to make it easier to follow the contents.
A: We have changed to “The Fv/Fm of the two families was not significantly different from that of the control under the 10℃ treatment. However, there was a significant difference between the treatment at 10℃R and the control. The Fv/Fm was significantly lower than that of the control under 0℃ and -5℃ stress. After the 0℃R treatment, the Fv/Fm of the two families increased. In contrast, the Fv/Fm decreased sharply under -5℃ stress (Figure 2)” (line 91-94)
Line 108: Delete “temperature decreased.”
A:Yes, we delete.
Line 126: italic
A: Yes, we did.
Line 162-167, 169-172, 172-177, 224-229, 229-236, 241-246: Please break up the sentence to improve readability. It's currently quite challenging to follow. Or rewrite It to make it easier to follow the contents. Using too many “and” in one sentence is not recommended. See the example below:
**Example for breaking up Line 229-236
“Under normal growth conditions, the cold-tolerant families displayed a large number of starch granules in their acinar cells. These granules were lighter in color compared to the control. Upon the 10℃ treatments, there was a significant decrease in the number of starch granules; however, their number sharply increased with the 0℃ treatment. Overall, the cold-tolerant families had more starch granules than the non-cold-tolerant families. Both the number and color of the granules were reduced under the 10℃ and 0℃ treatments. No starch granules were detected in the acinar cells of either family following the -5℃ treatment (Figure 4)”
A: We have broke up the long sentences into short sentences,which will make readers easily to follow.(line117-121,line123-126,line13-130,line157-160,line160-166,line 170-174)
Line 200, 201: Italic
A: Yes, we did.
Discussion:
The discussion section could be strengthened by delving deeper into the physiological mechanisms underlying the observed photosynthetic changes under cold stress. A comparative analysis highlighting the differences between the cold-tolerant and cold-sensitive families would clarify how specific adaptations confer cold tolerance. Additionally, broadening the scope to include ecological implications and practical applications, such as implications for forestry management and plant breeding, would make the findings more relevant and impactful. Enriching the discussion in these ways would provide a clearer understanding of the study’s broader significance.
- The manuscript mentions general trends in response to cold temperatures but could benefit from a more detailed comparison between the cold-tolerant and cold-sensitive families. Specifically, how do these families differ not just in the magnitude of change in photosynthetic parameters but also in their physiological responses and recovery capabilities? Highlighting these differences could provide valuable insights into the mechanisms of cold tolerance.
- The discussion on the reversibility of the damage at -5°C and the capacity for repair at temperatures above 0°C is crucial. However, it might be useful to discuss the long-term impacts of such stress if it were to occur repeatedly or persist over a longer period. How resilient are these plants to recurrent or prolonged cold stress?
- Extending the discussion to include ecological or evolutionary perspectives could enrich the analysis. How do these physiological responses to cold stress influence the ecological distribution and evolutionary adaptations of massoniana? How might these traits affect the plant's competitive ability and survival in natural settings under climate change scenarios?
A:The expert provided excellent suggestions for discussion. Based on the current discussion, we followed the expert's advice to conduct a more results-oriented and comprehensive discussion. Sentences with red color are the contents we added. (line 177-180,182-183,184-185.196-203,208-210, 231-243,260-266)
Materials and Methods:
The materials and Methods section is missing many important information. By reading this section we should be able to redo your experiment. So please add more detailed information.
A: We have tried to improve the presentation of the Materials and Methods section, and have made it possible for the reader to achieve the effect of repeating the experiment.
Additionally, I understand that 25℃ is for Control but I could not understand why the -5 ℃ treatment does not have -5 ℃R. Can you add a reason for that?
A:Here is an analysis of this very good question: Since we used 30-day-old seedlings, as shown in Figure 1, although the cold-resistant family exhibited relatively strong cold resistance under -5℃ treatment, neither family was able to recover to normal growth levels during the warming process. However, we are certain that this result is related to the size of the seedlings we treated and does not imply that Masson pine saplings (over one year old) cannot withstand -5℃ low temperatures. Therefore, under the same treatment conditions, there are no samples of -5℃R. Our previous field experiments demonstrated that recovery is possible after -5℃ treatment (photos from January 2021) by April 2022 (as shown in the figure below). More direct evidence is that in Figure 1, the -5℃R images clearly show this, but at the time, it was believed that further experiments could not be conducted, so no photos were taken and preserved.
4.1. Design and measurement of greenhouse experiments
While the section provides a broad overview of the experimental setup, it could benefit significantly from a more detailed description of the experimental design and the specific treatments applied. It appears that six treatment groups were established, but the description of how these groups differ and the specific parameters of each treatment group are not clearly articulated. In the section title, it says “greenhouse experiments” but I could not find when the seedlings were moved to the greenhouse. Additionally, the experimental design is not adequately described. This information is crucial for understanding the experimental framework and ensuring the reproducibility of the results.
A:We agree with the reviewer and add more details of the work.(line275-279) (line280-281)
4.2.1. Chlorophyll fluorescence parameters determination
I noticed several areas where typos and redundant phrases in the section. To enhance the readability and accuracy, I recommend a thorough review to correct these typos. Additionally, simplifying sentences to remove unnecessary repetition will strengthen this section. The manuscript mentions the use of the FMS-2 portable pulse-modulated fluorometer twice in a way that seems redundant. I recommend consolidating these mentions into a single, clear description of the equipment used and its specific application in the context of your study.
A: We have modified 4.2.1.Delete the second FMS-2 and add more details. (line 295-306)
Light-adapted measurements were not written here. Add all the parameters you measured using FMS-2.
The method of calculating indices such as ??, ??/??, ФPSII, and NPQ should be explicitly stated, including any formulas used. This detail is crucial for replicability and for ensuring that other researchers can accurately compare your results with findings from other studies.
Please confirm and describe the sampling strategy used (e.g., number of leaves measured per plant, number of plants per treatment group) to ascertain the statistical validity of the results. This information is vital for assessing the robustness of your study's conclusions.
A:We have added the Light-adapted measurements method. ??, ??/??, ФPSII, and NPQ have been explained explicitly. Formulas have been added. Mealwhile, number of leaves measured per plant, number of plants per treatment group have been added.(line 309-317)
4.2.4. Microscopic observation
In the methodology section, it's crucial to maintain clarity and organization to ensure that readers can follow the procedures effectively. While the description you provided covers the steps involved in the experimental process, the repeated use of "and" makes the sentence long and somewhat cumbersome. Consider breaking down each step into separate sentences or using bullet points to list the actions taken. This will improve readability and make it easier for readers to understand and replicate the methodology. Additionally, providing context or explaining the purpose of each action can enhance the overall understanding of the experimental procedures. For instance, it would be beneficial to provide additional information or context regarding the "solid green-iodine potassium iodide staining" mentioned in the methodology. This could include details about the purpose of this staining technique, how it is performed, and its significance in the experimental process. Providing such information not only enhances the completeness of your methodology section but also helps readers understand the specific techniques and procedures used in your study.
A: We have added more details of the methodology (line 325-338)
4.3. Data analysis
The current description of the data analysis primarily outlines the software tools utilized but does not detail the checks for fundamental statistical assumptions required for ANOVA. It is essential to verify that the data meet the assumptions of normality and homogeneity of variances before conducting ANOVA, as deviations from these assumptions can lead to incorrect conclusions. If you have already performed these tests, please include a mention of this in the methodology section, specifying which tests were used and any subsequent steps taken to address potential issues.
A: We have added the steps of normal distribution before conducting ANOVA.(line347-350)
Other minor comments:
Line 338: Move (W: H) to after “high”
A: We have moved it into the right place.(line 274)
Line 340: What do you mean by normally?
A:we have added standard to explain. “According to the technical regulations for P. massoniana seedling cultivation (standard number DB45/T 2384-2021)”is the new content.(line275-277)
Line 351: Please provide an explanation for morphological observation. It is difficult to visualize how did you evaluate it.
A:We conducted direct observations for morphological characteristics, and we believe they are quite straightforward. In fact, we collected data on wilting degree during the experiment. If the experts deem it necessary, we can upload this as supplementary figures.
Line 358: Provide the location of the product. I assume this is the right one “(Hansatech, Norfolk, UK)” but please check.
A: We have added the product information. (line 295)
Line 363: delete “The” (not only “The” but also the entire paragraph has issues. Please re-rite this paragraph.
A: We deleted “the” and have revised the paragraph. (line 295-306)
Line 364 Descriptions for Fo’, Fm’, are not correct.
A: We have revised the descriptions. (line295-306)
Line 365: You mean “(ФPSII)”?
A: We have changed it. (line 302)
Line 387: Which data was processed? If all data were, add “All data”.
A: We have added “all” in front of data (line 347)
Line 387: I believe you did two-way ANOVA if so, change “ANOVA” to “two-way analysis of variance (ANOVA)”
A: Yes, We have change it into two-way analysis of variance (ANOVA)(line 347)
Conclusion:
Based on the changes you will make in the discussion, please revise the conclusion as well.
Other minor comments:
Line 390: Italic for scientific name
Yes, we did.(line 352)
**Please check all the scientific names in the manuscript and use italics for all.
Yes, we did.

Reviewer 2 Report
Comments and Suggestions for Authors
The reviewed work concerns the impact of low-temperature stress on the physiological character and microstructure of Pinus massoniana stems and leaves.
The aims of the article were stated in the introduction. The research methods are appropriate and clearly described. The structure of the work (its composition and division into chapters) is correct. Tables and figures are relevant to the main text and legible. The abstract presents the aim, methods and results of the article. The summary summarizes the main results and contributions to the current state of knowledge. The language and terms used are correct. However, some suggestions should be considered:
1. Line 82. Results. I think the sentence: “This section may be divided by subheadings. "It should provide a concise and precise description of the experimental results, their interpretation, as well as the experimental conclusions that can be drawn" is not needed here.
2. Fig 2. The drawing shows the letters: a, b, c…. I think the authors should explain what they mean.
3. Line 358: please provide more details about the equipment.
4. Line: 386: I think it is necessary to explain what statistical analyzes were used.
5. Literature. The authors cited 36 sources. However, this is not the latest literature. Only 11 items are articles from the last 5 years. Please cite newer items.
I have no hesitation in recommending publication with minor changes.

Author Response
Response to reviewer
Thank you very much for your comments.
- Line 82. Results. I think the sentence: “This section may be divided by subheadings. "It should provide a concise and precise description of the experimental results, their interpretation, as well as the experimental conclusions that can be drawn" is not needed here.
A: Yes, we have deleted these contents
- Fig 2. The drawing shows the letters: a, b, c…. I think the authors should explain what they mean.
A: We have added the content “Different lowercase letters indicat significant differences between different families under the same temperature (P<0.05).” (line 478-479)
- Line 358: please provide more details about the equipment.
A: Yes, we changed to “(Hansatech, Norfolk, UK)”(line 295)
- Line: 386: I think it is necessary to explain what statistical analyzes were used.
A:Yes, We changed to “All data were processed using Excel, analysed by two-way analysis of variance (ANOVA) using SPSS 13.0 software and plotted using GraphPad Prism 7.0. The normality distribution was assessed using the Kolmogorov-Smirnov test (K-S test) in SPSS. If the P-value is greater than the set significance level (0.05), the data is considered to follow a normal distribution.”(line347-350)
- The authors cited 36 sources. However, this is not the latest literature. Only 11 items are articles from the last 5 years. Please cite newer items.
A: We have added some latest new references. eg.
- Guan, Y.L.; Hwarari, D.; Korboe H.M.; Ahmad, B.; Cao, Y.W.; Movahedi, A.; Yang, L.M. Low temperature stress-induced perception and molecular signaling pathways in plants. Environmental and Experimental Botany 2023, 207: 105190.
- Kidokoro, S.; Shinozaki, K.; Yamaguchi-Shinozaki, K. Transcriptional regulatory network of plant cold-stress responses. Trends in Plant Science 2022, 27: 922–935.
15.Cano-Ramirez, D.L. L.; Panter, P.E.E.; Takemura, T., Tara Saskia de Fraine , Luíza Lane de Barros Dantas, Richard Dekeya, Thiago Barros-Galvão, Pirita Paajanen, Annalisa Bellandi, Tom Batstone, Bethan F. Manley, Kan Tanaka, Sousuke Imamura, Keara A. Franklin, Heather Knight & Antony N. Dodd. Low-temperature and circadian signals are integrated by the sigma factor SIG5. Nature Plants 2023.

Round 2
Reviewer 1 Report
Comments and Suggestions for Authors
The manuscript “Effects of Low Temperature Stress on Physiological Characteristics and Microstructure of Stems and Leaves of Pinus massoniana L” has been significantly improved.
Here are some minor comments:
Abstract: add “one of” before “the most important conifer species”
Line 17-20: Change “30-day-old seedlings from several cold-tolerant families were used as materials to simulate the morphological traits of needles and stems, chlorophyll fluorescence characteristics in an artificial climate chamber, and the changes in protective enzymes, starch, and lignin under different low-temperature stresses.” to “This study used 30-day-old seedlings from various cold-tolerant families to examine the morphological traits of needles and stems, chlorophyll fluorescence characteristics, protective enzymes, and changes in starch and lignin under different low-temperature stresses in an artificial climate chamber.”. If you are agreeing with this.
Line 20: delete “,” after “that”
Line 48: delete “. Plants produce” and add “and by producing”
Line 49: delete “which endow plants with a variety of osmotic regulatory capabilities.” And add “which enhance their osmotic regulatory capabilities.”
Line 61: Change “occurrence of extreme events” to “increasing frequency of extreme weather events”.
Line 61: Delete “subjected to numerous rain, snow, freezing, and low-temperature disasters,” Here, probably the authors forgot to delete the former sentence??
Line 107 Figure 2: Put “Cold-tolerant” and “Cold-sensitive” to the figures instead of 20-628 and 20-654.
Line 109: Change “indicat” to “indicate”
Line 324-332: Make all past tense
Figure 4: Explain what the red arrow indicates in the figure caption
Figure 5, 6: Please put arrows like other Figures so that the readers would immediately know where to look at.
Line 296-297: add species name
Line 301: delete (W:H)
Line 303: Italic “P. massoniana”
Line 313: Change “handled” to “maintained”
Line 315-318: Move this to after “4.2 Experimental measurements”
Line 295: Change “4.1 Design and measurement of greenhouse experiments” to “4.1 Plant Materials and Environmental Controls”
Line 378: Why suddenly talk about “Pinus sylvestris”?
Line 378-383: The sentence is too long. It is very hard to read.
Change “The cold-tolerant and non-cold-tolerant families of Pinus sylvestris have similar trends in phenotypic morphology, physiology and biochemistry, and tissue structure changes under low temperature stress, and respond to low temperature stress by increasing ФPSII and ETR indexes, enhancing membrane permeability, improving the activity of protective enzymes, enhancing osmotic regulating substances, and increasing the level of starch grains and the content of lignin in the xylem, and the cold-tolerant families can accumulate more nutrients and metabolites to improve cold tolerance, thus demonstrating cold tolerance.” to “The cold-tolerant and non-cold-tolerant families of Pinus sylvestris exhibited similar trends in phenotypic morphology, physiology, biochemistry, and tissue structure changes under low-temperature stress. They responded to the stress by increasing ФPSII and ETR indexes, enhancing membrane permeability, improving the activity of protective enzymes, boosting osmotic regulating substances, and increasing the levels of starch grains and lignin content in the xylem. Cold-tolerant families accumulated more nutrients and metabolites, thereby demonstrating improved cold tolerance.”
Ensure the species in red avobe.
Comments on the Quality of English Language
Minor editing of English language required
Author Response
Abstract: add “one of” before “the most important conifer species”
A: We have revised it and added “one of” before “the most important conifer species”.
Line 17-20: Change “30-day-old seedlings from several cold-tolerant families were used as materials to simulate the morphological traits of needles and stems, chlorophyll fluorescence characteristics in an artificial climate chamber, and the changes in protective enzymes, starch, and lignin under different low-temperature stresses.” to “This study used 30-day-old seedlings from various cold-tolerant families to examine the morphological traits of needles and stems, chlorophyll fluorescence characteristics, protective enzymes, and changes in starch and lignin under different low-temperature stresses in an artificial climate chamber.”. If you are agreeing with this.
A: Yes, we agreed with it and changed it.
Line 20: delete “,” after “that”
A: Yes, we did.
Line 48: delete “. Plants produce” and add “and by producing”
A:Yes, we deleted.
Line 49: delete “which endow plants with a variety of osmotic regulatory capabilities.” And add “which enhance their osmotic regulatory capabilities.”
A:Yes, we deleted and added “which enhance their osmotic regulatory capabilities.”
Line 61: Change “occurrence of extreme events” to “increasing frequency of extreme weather events”.
A:Yes, we changed
Line 61: Delete “subjected to numerous rain, snow, freezing, and low-temperature disasters,” Here, probably the authors forgot to delete the former sentence?
A:Yes, we deleted this time.
Line 107 Figure 2: Put “Cold-tolerant” and “Cold-sensitive” to the figures instead of 20-628 and 20-654.
A: We have changed the Figure.
Line 109: Change “indicat” to “indicate”
A:Yes, this is a mistake and we have changed it
Line 324-332: Make all past tense
A: We have changed the tense to past tense.
Figure 4: Explain what the red arrow indicates in the figure caption
A: Yes, we have added the interpretation.
Figure 5, 6: Please put arrows like other Figures so that the readers would immediately know where to look at.
A: Yes, we have added the arrows and interpretations.
Line 296-297: add species name
A:Yes, we did.
Line 301: delete (W:H)
A:Yes, we deleted.
Line 303: Italic “P. massoniana”
A:Yes, we did.
Line 313: Change “handled” to “maintained”
A:Yes, we did.
Line 315-318: Move this to after “4.2 Experimental measurements”
A: Yes,we agreed and moved it after “4.2 Experimental measurements”
